# Variational Inverse Control with Events: A General Framework for Data-Driven Reward Definition

**Justin Fu**[*]  **Avi Singh**[*]  **Dibya Ghosh**  **Larry Yang**  **Sergey Levine**

University of California, Berkeley

{justinfu, avisingh, dibyaghosh, larrywyang, svlevine}@berkeley.edu

## Abstract

The design of a reward function often poses a major practical challenge to real-world applications of reinforcement learning. Approaches such as inverse reinforcement learning attempt to overcome this challenge, but require expert demonstrations, which can be difficult or expensive to obtain in practice. We propose variational inverse control with events (VICE), which generalizes inverse reinforcement learning methods to cases where full demonstrations are not needed, such as when only samples of desired goal states are available. Our method is grounded in an alternative perspective on control and reinforcement learning, where an agent's goal is to maximize the probability that one or more events will happen at some point in the future, rather than maximizing cumulative rewards. We demonstrate the effectiveness of our methods on continuous control tasks, with a focus on high-dimensional observations like images where rewards are hard or even impossible to specify.

## 1  Introduction

Reinforcement learning (RL) has shown remarkable promise in recent years, with results on a range of complex tasks such as robotic control (Levine et al., 2016) and playing video games (Mnih et al., 2015) from raw sensory input. RL algorithms solve these problems by learning a policy that maximizes a reward function that is considered as part of the problem formulation. There is little practical guidance that is provided in the theory of RL about how these rewards should be designed. However, the design of the reward function is in practice critical for good results, and reward misspecification can easily cause unintended behavior (Amodei et al., 2016). For example, a vacuum cleaner robot rewarded to pick up dirt could exploit the reward by repeatedly dumping dirt on the ground and picking it up again (Russell & Norvig, 2003). Additionally, it is often difficult to write down a reward function at all. For example, when learning policies from high-dimensional visual observations, practitioners often resort to using motion capture (Peng et al., 2017) or specialized computer vision systems (Rusu et al., 2017) to obtain rewards.

As an alternative to reward specification, imitation learning (Argall et al., 2009) and inverse reinforcement learning (Ng & Russell, 2000) instead seek to mimic expert behavior. However, such approaches require an expert to show *how* to solve a task. We instead propose a novel problem formulation, variational inverse control with events (VICE), which generalizes inverse reinforcement learning to alternative forms of expert supervision. In particular, we consider cases when we have examples of a desired final outcome, rather than full demonstrations, so the expert only needs to show *what* the desired outcome of a task is (see Figure 1). A straightforward way to make use of these desired outcomes is to train a classifier (Pinto & Gupta, 2016; Tung et al., 2018) to distinguish desired and undesired states. However, for these approaches it is unclear how to correctly sample negatives and whether using this classifier as a reward will result in the intended behavior, since an

---

[*]equal contribution

RL agent can learn to exploit the classifier, in the same way it can exploit human-designed rewards. Our framework provides a more principled approach, where classifier training corresponds to learning probabilistic graphical model parameters (see Figure 2), and policy optimization corresponds to inferring the optimal actions. By selecting an inference query which corresponds to our intentions, we can mitigate reward hacking scenarios similar to those previously described, and also specify the task with examples rather than manual engineering.

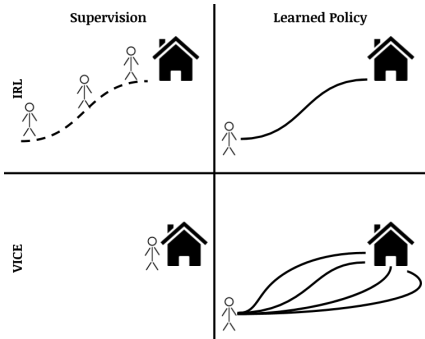

Our inverse formulation is based on a corresponding forward control framework which reframes control as inference in a graphical model. Our framework resembles prior work (Kappen et al., 2009; Toussaint, 2009; Rawlik et al., 2012), but we extend this connection by replacing the conventional notion of rewards with event occurence variables. Rewards correspond to log-probabilities of events, and value functions can be interpreted as backward messages that represent log-probabilities of those events occurring. This framework retains the full expressivity of RL, since any rewards can be expressed as log-probabilities, while providing more intuitive guidance on task specification. It further allows us to express various intentions, such as for an event to happen at least once, exactly once at any time step, or once at a specific timestep. Crucially, our framework does not require the agent to *observe* the event happening, but only to know the probability that it occurred. While this may seem unusual, it is more practical in the real world, where success may be determined by probabilistic models that themselves carry uncertainty. For example, the previously mentioned vacuum cleaner robot needs to estimate from its observations whether its task has been accomplished and would never receive direct feedback from the real world whether a room is clean.

Figure 1: Standard IRL requires full expert demonstrations and aims to produce an agent that mimics the expert. VICE generalizes IRL to cases where we only observe final desired outcomes, which does not require the expert to actually know *how* to perform the task.

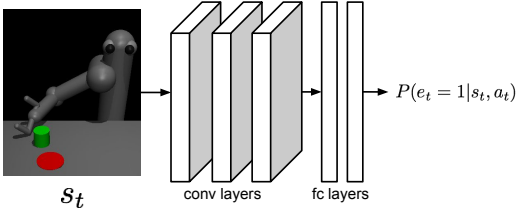

Our contributions are as follows. We first introduce the event-based control framework by extending previous control as inference work to alternative queries which we believe to be useful in practice. This view on control can ease the process of reward engineering by mapping a user's intention to a corresponding inference query in a probabilistic graphical model. Our experiments demonstrate how different queries can result in different behaviors which align with the corresponding intentions. We then propose methods to learn event probabilities from data, in a manner analogous to inverse reinforcement learning. This corresponds to the use case where designing event probabilities by hand is difficult, but observations (e.g., images) of successful task completion are easier to provide. This approach is substantially easier to apply in practical situations, since full demonstrations are not required. Our experiments demonstrate that our framework can be used in this fashion for policy learning from high dimensional visual observations where rewards are hard to specify. Moreover, our method substantially outperforms baselines such as sparse reward RL, indicating that our framework provides an automated shaping effect when learning events, making it feasible to solve otherwise hard tasks.

Figure 2: Our framework learns event probabilities from data. We use neural networks as function approximators to model this distribution, which allows us to work with high dimensional observations like images.

## 2 Related work

Our reformulation of RL is based on the connection between control and inference (Kappen et al., 2009; Ziebart, 2010; Rawlik et al., 2012). The resulting problem is sometimes referred to as maximum entropy reinforcement learning, or KL control. Duality between control and inference in the case of linear dynamical systems has been studied in Kalman (1960); Todorov (2008). Maximum entropy objectives can be optimized efficiently and exactly in linearly solvable MDPs (Todorov, 2007) and

environments with discrete states. In linear-quadratic systems, control as inference techniques have been applied to solve path planning problems for robotics (Toussaint, 2009). In the context of deep RL, maximum entropy objectives have been used to derive soft variants of Q-learning and policy gradient algorithms (Haarnoja et al., 2017; Schulman et al., 2017; O'Donoghue et al., 2016; Nachum et al., 2017). These methods embed the standard RL objective, formulated in terms of rewards, into the framework of probabilistic inference. In contrast, we aim specifically to reformulate RL in a way that does not require specifying arbitrary scalar-valued reward functions.

In addition to studying inference problems in a control setting, we also study the problem of learning event probabilities in these models. This is related to prior work on inverse reinforcement learning (IRL), which has also sought to cast learning of objectives into the framework of probabilistic models (Ziebart et al., 2008; Ziebart, 2010). As explained in Section 5, our work generalizes IRL to cases where we only provide examples of a desired outcome or goal, which is significantly easier to provide in practice since we do not need to know how to achieve the goal.

Reward design is crucial for obtaining the desired behavior from RL agents (Amodei et al., 2016). Ng & Russell (2000) showed that rewards can be modified, or shaped, to speed up learning without changing the optimal policy. Singh et al. (2010) study the problem of optimal reward design, and introduce the concept of a fitness function. They observe that a proxy reward that is distinct from the fitness function might be optimal under certain settings, and Sorg et al. (2010) study the problem of how this optimal proxy reward can be selected. Hadfield-Menell et al. (2017) introduce the problem of inferring the true objective based on the given reward and MDP. Our framework aids task specification by introducing two decisions: the selection of the inference query that is of interest (i.e., when and how many times should the agent cause the event?), and the specification of the event of interest. Moreover, as discussed in Section 6, we observe that our method automatically provides a reward shaping effect, allowing us to solve otherwise hard tasks.

## 3 Preliminaries

In this section we introduce our notation and summarize how control can be framed as inference. Reinforcement learning operates on Markov decision processes (MDP), defined by the tuple $(\mathcal{S}, \mathcal{A}, \mathcal{T}, r, \gamma, \rho_0)$. $\mathcal{S}, \mathcal{A}$ are the state and action spaces, respectively, $r$ is a reward function, which is typically taken to be a scalar field on $\mathcal{S} \times \mathcal{A}$, and $\gamma \in (0, 1)$ is the discount factor. $\mathcal{T}$ and $\rho_0$ represent the dynamics and initial state distributions, respectively.

### 3.1 Control as inference

In order to cast control as an inference problem, we begin with the standard graphical model for an MDP, which consists of states and actions. We incorporate the notion of a goal with an additional variable $e_t$ that depends on the state (and possibly also the action) at time step $t$, according to $p(e_t|s_t, a_t)$. If the goal is specified with a reward function, we can define $p(e_t = 1|s_t, a_t) = e^{r(s,a)}$ which, as we discuss below, leads to a maximum entropy version of the standard RL framework. This requires the rewards to be negative, which is not restrictive in practice, since if the rewards are bounded we can re-center them so that the maximum value is 0. The structure of this model is presented in Figure 3, and is also considered in prior work, as discussed in the previous section.

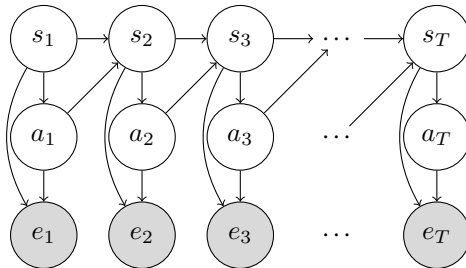

Figure 3: A graphical model framework for control. In maximum entropy reinforcement learning, we observe $e_{1:T} = 1$ and can perform inference on the trajectory to obtain a policy.

The maximum entropy reinforcement learning objective emerges when we condition on $e_{1:T} = 1$. Consider computing a backward message $\beta(s_t, a_t) = p(e_{t:T} = 1|s_t, a_t)$. Letting $Q(s_t, a_t) = \log \beta(s_t, a_t)$, notice that the backward messages encode the backup equations

$$Q(s_t, a_t) = r(s_t, a_t) + \log E_{s_{t+1}}[e^{V(s_{t+1})}] \qquad V(s_t) = \log \int_{a \in \mathcal{A}} e^{Q(s_t, a)} da \ .$$

We include the full derivation in Appendix A, which resembles derivations discussed in prior work (Ziebart et al., 2008). This backup equation corresponds to maximum entropy RL, and is equivalent to soft Q-learning and causal entropy RL formulations in the special case of deterministic dynamics (Haarnoja et al., 2017; Schulman et al., 2017). For the case of stochastic dynamics, maximum-entropy RL is optimistic with respect to the dynamics and produces risk-seeking behavior, and we refer the reader to Appendix B, which covers a variational derivation of the policy objective which properly handles stochastic dynamics.

# 4   Event-based control

In control as inference, we chose $\log p(e_t = 1|s_t, a_t) = r(s, a)$ so that the resulting inference problem matches the maximum entropy reinforcement learning objective. However, we might also ask: what does the variable $e_t$, and its probability, represent? The connection to graphical models lets us interpret rewards as the log-probability that an event occurs, and the standard approach to reward design can also be viewed as specifying the probability of some binary event, that we might call an optimality event. This provides us with an alternative way to think about task specification: rather than using arbitrary scalar fields as rewards, we can specify the events for which we would like to maximize the probability of occurrence.

We now outline inference procedures for different types of problems of interest in the graphical model depicted in Figure 3. In Section 5, we will discuss learning procedures in this graphical model which allow us to specify objectives from data. The strength of the events framework for task specification lies in both its intuitive interpretation and flexibility: though we can obtain similar behavior in standard reinforcement learning, it may require considerable reward tuning and changes to the overall problem statement, including the dynamics. In contrast, events provides a single unified framework where the problem parameters remain unchanged, and we simply ask the appropriate queries. We will discuss:

- **ALL** query: $p(\tau|e_{1:T} = 1)$, meaning the event should happen at each time step.
- **AT** query: $p(\tau|e_{t^*} = 1)$, meaning the event should happen at a specific time $t^*$.
- **ANY** query: $p(\tau|e_1 = 1 \text{ or } e_2 = 1 \text{ or } ... e_T = 1)$ meaning the event should happen on at least one time step during each trial.

We present two derivations for each query: a conceptually simple one based on maximum entropy and message passing (see Section 3.1), and one based on variational inference, (see Appendix B), which is more appropriate for stochastic dynamics. The resulting variational objective is of the form:

$$J(\pi) = -D_{KL}(\pi(\tau)||p(\tau|\text{evidence})) = E_{s_{1:T}, a_{1:T} \sim q}[\hat{Q}(s_{1:T}, a_{1:T}) + H^\pi(\cdot|s_{1:T})],$$

where $\hat{Q}$ is an empirical Q-value estimator for a trajectory and $H^\pi(\cdot|s_{1:T}) = -\sum_{t=0}^{T} \log \pi(a_t|s_t)$ represents the entropy of the policy. This form of the objective can be used in policy gradient algorithms, and in special cases can also be written as a recursive backup equation for dynamic programming algorithms. We directly present our results here, and present more detailed derivations (including extensions to discounted cases) in Appendices C and D.

## 4.1   ALL and AT queries

We begin by reviewing the ALL query, when we wish for an agent to trigger an event at every timestep. This can be useful, for example, when expressing some continuous task such as maintaining some sort of configuration (such as balancing on a unicycle) or avoiding an adverse outcome, such as not causing an autonomous car to collide. As covered in Section 3.1, conditioning on the event at all time steps mathematically corresponds to the same problem as entropy maximizing RL, with the reward given by $\log p(e_t = 1|s_t, a_t)$.

**Theorem 4.1** (ALL query). *In the ALL query, the message passing update for the Q-value can be written as:*

$$Q(s_t, a_t) = \log p(e_t = 1|s_t, a_t) + \log E_{s_{t+1}}[e^{V(s_{t+1})}],$$

*where $Q(s_t, a_t)$ represents the log-message $\log p(e_{t:T} = 1|s_t, a_t)$. The corresponding empirical Q-value can be written recursively as:*

$$\hat{Q}(s_{t:T}, a_{t:T}) = \log p(e_t = 1|s_t, a_t) + \hat{Q}(s_{t+1:T}, a_{t+1:T}).$$

*Proof.* See Appendices C.1 and  D.1 ☐

The AT query, or querying for the event at a specific time step, results in the same equations, except $\log p(e = 1|s_t, a_t)$, is only given at the specified time $t^*$. While we generally believe that the ANY query presented in the following section will be more broadly applicable, there may be scenarios where an agent needs to be in a particular configuration or location at the end of an episode. In these cases, the AT query would be the most appropriate.

## 4.2   ANY query

The ANY query specifies that an event should happen at least once before the end of an episode, without regard for when in particular it takes place. Unlike the ALL and AT queries, the ANY query does not correspond to entropy maximizing RL and requires a new backup equation. It is also in many cases more appropriate: if we would like an agent to accomplish some goal, we might not care when in particular that goal is accomplished, and we likely don't need it to accomplish it more than once. This query can be useful for specifying behaviors such as reaching a goal state, completion of a task, etc. Let the stopping time $t^* = \min\{t \geq 0 | e_t = 1\}$ denote the first time that the event occurs.

**Theorem 4.2** (ANY query). *In the ANY query, the message passing update for the Q-value can be written as:*

$$Q(s_t, a_t) = \log \left( p(e_t = 1|s_t, a_t) + p(e_t = 0|s_t, a_t) E_{s_{t+1}}[e^{V(s_{t+1})}] \right)$$

*where $Q(s_t, a_t)$ represents the log-message $\log p(t \leq t^* \leq T|s_t, a_t)$. The corresponding empirical Q-value can be written recursively as:*

$$\hat{Q}(s_{t:T}, a_{t:T}) = \log \left( p(e_t = 1|s_t, a_t) + p(e_t = 0|s_t, a_t) e^{\hat{Q}(s_{t+1:T}, a_{t+1:T})} \right).$$

*Proof.* See Appendices C.2 and  D.2 ☐

This query is related to first-exit RL problems, where an agent receives a reward of 1 when a specified goal is reached and is immediately moved to an absorbing state but it does not require the event to actually be observed, which makes it applicable to a variety of real-world situations that have uncertainty over the goal. The backup equations of the ANY query are equivalent to the first-exit problem when $p(e|s, a)$ is deterministic. This can be seen by setting $p(e = 1|s, a) = r_F(s, a)$, where $r_F(s, a)$ is an goal indicator function that denotes the reward of the first-exit problem. In this case, we have $Q(s, a) = 0$ if the goal is reachable, and $Q(s, a) = -\infty$ if not. In the first-exit case, we have $Q(s, a) = 1$ if the goal is reachable and $Q(s, a) = 0$ if not - both cases result in the same policy.

## 4.3   Sample-based optimization using policy gradients

In small, discrete settings with known dynamics, we can use the backup equations in the previous section to solve for optimal policies with dynamic programming. For large problems with unknown dynamics, we can also derive model-free analogues to these methods, and apply them to complex tasks with high-dimensional function approximators. One commonly used method is the policy gradient, and which we can derive via logarithmic differentiation as:

$$\nabla_\theta J(\theta) = -\nabla_\theta D_{KL}(\pi_\theta(\tau) || p(\tau|\text{evidence}))$$

$$= E_{s_{1:T}, a_{1:T} \sim \pi_\theta} \left[ \sum_{t=1}^{T} \nabla \log \pi_\theta(a_t|s_t)(\hat{Q}(s_{1:T}, a_{1:T}) + H^\pi(\cdot|s_{t:T})) \right]$$

Under certain assumptions we can replace $\hat{Q}(s_{1:T}, a_{1:T})$ with $\hat{Q}(s_{t:T}, a_{t:T})$ to obtain an estimator which only depends on future returns. See Appendix E for further explanation.

This estimator can be integrated into standard policy gradient algorithms, such as TRPO Schulman et al. (2015), to train expressive inference models using neural networks. Extensions of our approach to other RL methods with function approximation, such as Q-learning and approximate dynamic programming, can also be derived from the backup equations, though this is outside the scope of the present work.

---

**Algorithm 1** VICE: Variational Inverse Control with Events

---

1: Obtain examples of expert states and actions $s_i^E, a_i^E$
2: Initialize policy $\pi$ and binary discriminator $D_\theta$.
3: **for** step $n$ in $\{1, \ldots, N\}$ **do**
4:     Collect states and actions $s_i = (s_1, ..., s_T), a_i = (a_1, ..., a_T)$ by executing $\pi$.
5:     Train $D_\theta$ via logistic regression to classify expert data $s_i^E, a_i^E$ from samples $s_i, a_i$.
6:     Update $\pi$ with respect to $p_\theta$ using the appropriate inference objective.
7: **end for**

---

## 5   Learning event probabilities from data

In the previous section, we presented a control framework that operates on events rather than reward functions, and discussed how the user can choose from among a variety of inference queries to obtain a desired outcome. However, the event probabilities must still be obtained in some way, and may be difficult to hand-engineer in many practical situations - for example, an image-based deep RL system may need an image classifier to determine if it has accomplished its goal. In such situations, we can ask the user to instead supply examples of states or observations where the event has happened, and learn the event probabilities $p_\theta(e = 1|s, a)$. Inverse reinforcement learning corresponds to the case when we assume the expert triggers an event at all timesteps (the ALL query), in which case we require full demonstrations. However, if we assume the expert is optimal under an ANY or AT query, full demonstrations are not required because the event is not assumed to be triggered at each timestep. This means our supervision can be of the form of a desired set of states rather than full trajectories. For example, in the vision-based robotics case, this means that we can specify goals using images of a desired goal state, which are much easier to obtain than full demonstrations.

Formally, for each query, we assume our dataset of states and actions $(s, a) \sim p_{data}(s, a|e = 1)$ when the event has happened, assuming the data-generating policy follows one of our inference queries. Our objective is imitation: we wish to train a model which produces samples that match the data. To that end, we learn the parameters of the model $p_\theta(s, a|e = 1)$, trained with the maximum likelihood objective:

$$\mathcal{L}(\theta) = -E_{p_{data}} \left[ \log p_\theta(s, a|e = 1) \right]$$

The gradient of this model is:

$$\nabla_\theta \mathcal{L}(\theta) = -E_{p_{data}} \left[ \nabla_\theta \log p_\theta(s, a|e = 1) \right] + E_{p_\theta} \left[ \nabla_\theta \log p_\theta(s, a|e = 1) \right] \quad (1)$$

Where the second term corresponds to the gradient of the partition function of $p_\theta(s, a|e = 1)$. Thus, this implies an algorithm where we sample states and actions from the model $p_\theta$ and use them to compute the gradient update.

### 5.1   Sample-based optimization with discriminators

In high-dimensional settings, a convenient method to perform the gradient update in Eqn. 1 is to embed the model $p_\theta(s, a|\text{evidence})$ within a discriminator between samples $p_\theta$ and data $p_{data}$ and take the gradient of the cross-entropy loss. Second, in order to draw samples from the model we instead train a "generator" policy via variational inference to draw samples from $p_\theta$. The variational inference procedure corresponds to those outlined in Section 4.

Specifically, we adapt the method of Fu et al. (2018), which alternates between training a discriminator with the fixed form

$$D_\theta(s, a) = p_\theta(s, a) / (p_\theta(s, a) + \pi(a|s))$$

to distinguish between policy samples and success states, and a policy that minimizes the KL divergence between $D_{KL}(\pi(s, a)||p_\theta(s, a| = 1))$. As shown in previous work (Finn et al., 2016b; Fu et al., 2018), the gradient of the cross entropy loss of the discriminator is equivalent to the gradient of Eqn. 1, and using the reward $\log D_\theta(s, a) - \log(1 - D_\theta(s, a))$ with the appropriate inference objective is equivalent to minimizing KL between the sampler and generator. We show the latter equivalence in Appendix F, and pseudocode for our algorithm is presented in Algorithm 1

# 6 Experimental evaluation

Our experimental evaluation aims to answer the following questions: (1) How does the behavior of an agent change depending on the choice of query? We study this question in the case where the event probabilities are already specified. (2) Does our event learning framework (VICE) outperform simple alternatives, such as offline classifier training, when learning event probabilities from data? We study this question in settings where it is difficult to manually specify a reward function, such as when the agent receives raw image observations. (3) Does learning event probabilities provide better shaped rewards than the ground truth event occurrence indicators? Additional videos and supplementary material are available at `https://sites.google.com/view/inverse-event`.

## 6.1 Inference with pre-specified event probabilities

We first demonstrate how the ANY and ALL queries in our framework result in different behaviors. We adapt TRPO (Schulman et al., 2015), a natural policy gradient algorithm, to train policies using our query procedures derived in Section 4. Our examples involve two goal-reaching domains, HalfCheetah and Lobber, shown in Figure 4. The goal of HalfCheetah is to navigate a 6-DoF

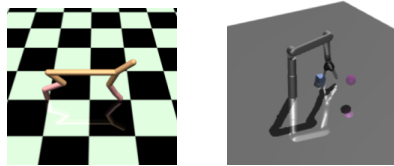

Figure 4: HalfCheetah and Lobber tasks.

agent to a goal position, and in Lobber, a robotic arm must throw a block to a goal position. To study the inference process in isolation, we manually design the event probabilities as $e^{-||x_{agent}-x_{target}||_2}$ for the HalfCheetah and $e^{-||x_{block}-x_{goal}||^2}$ for the Lobber.

The experimental results are shown in Table 1. While the average distance to the goal for both queries was roughly the same, the ANY query results in a much closer minimum distance. This makes sense, since in the ALL query the agent is punished for every time step it is not near the goal. The ANY query can afford to receive lower cumulative returns and instead has max-seeking behavior which more accurately reaches the target. Here, the ANY query better expresses our intention of reaching a target.

| Query | Avg. Dist | Min. Dist |
|---|---|---|
| HalfCheetah-ANY | 1.35 (0.20) | **0.97** (0.46) |
| HalfCheetah-ALL | **1.33** (0.16) | 2.01 (0.48) |
| HalfCheetah-Random | 8.95 (5.37) | 5.41 (2.67) |
| Lobber-ANY | 0.61 (0.12) | **0.25** (0.20) |
| Lobber-ALL | **0.59** (0.11) | 0.36 (0.21) |
| Lobber-Random | 0.93 (0.01) | 0.91 (0.01) |

Table 1: Results on HalfCheetah and Lobber tasks (5 trials). The ALL query generally results in superior returns, but the ANY query results in the agent reaching the target more accurately. Random refers to a random gaussian policy.

## 6.2 Learning event probabilities

We now compare our event probability learning framework, which we call variational inverse control with events (VICE), against an offline classifier training baseline. We also compare our method to learning from true binary event indicators, to see if our method can provide some reward shaping benefits to speed up the learning process. The data for learning event probabilities comes from success states. That is, we have access to a set of states $\{s_i^E\}_{i=1...n}$, which may

Table 2: Results on Maze, Ant and Pusher environments (5 trials). The metric reported is the final distance to the goal state (lower is better). VICE performs better than the classifier-based setup on all the tasks, and the performance is substantially better for the Ant and Pusher task. Detailed learning curves are provided in Appendix G.

| | Query type | Classifier | VICE (ours) | True Binary |
|---|---|---|---|---|
| **Maze** | ALL | 0.35 (0.29) | **0.20** (0.19) | 0.11 (0.01) |
| | ANY | 0.37 (0.21) | 0.23 (0.15) | |
| **Ant** | ALL | 2.71 (0.75) | 0.64 (0.32) | 1.61 (1.35) |
| | ANY | 3.93 (1.56) | **0.62** (0.55) | |
| **Push** | ALL | 0.25 (0.01) | **0.09** (0.01) | 0.17 (0.03) |
| | ANY | 0.25 (0.01) | 0.11 (0.01) | |

have been provided by the user, for which we know the event took place. This setting generalizes IRL, where instead of entire expert demonstrations, we simply have examples of successful states. The offline classifier baseline trains a neural network to distinguish success state ("positives") from states collected by a random policy. The number of positives and negatives in this procedure is kept balanced. This baseline is a reasonable and straightforward method to specify rewards in the standard RL framework, and provides a natural point of comparison to our approach, which can also be viewed as learning a classifier, but within the principled framework of control as inference. We evaluate these methods on the following tasks:

**Maze from pixels.** In this task, a point mass needs to navigate to a goal location through a small maze, depicted in Figure 5. The observations consist of 64x64 RGB images that correspond to an overhead view of the maze. The action space consists of X and Y forces on the robot. We use CNNs to represent the policy and the event distributions, training with 1000 success states as supervision.

**Ant.** In this task, a quadrupedal "ant" (shown in Figure 5) needs to crawl to a goal location, placed 3m away from its starting position. The state space contains joint angles and XYZ-coordinates of the ant. The action space corresponds to joint torques. We use 500 success states as supervision.

**Pusher from pixels.** In this task, a 7-DoF robotic arm (shown in Figure 5) must push a cylinder object to a goal location. The state space contains joint angles, joint velocities and 64x64 RGB images, and the action space corresponds to joint torques. We use 10K success states as supervision.

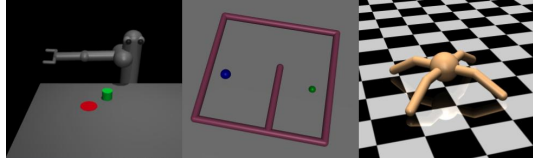

Training details and neural net architectures can be found in Appendix G. We also compare our method against a reinforcement learning baseline that has access to the true binary event indicator. For all the tasks, we define a "goal region", and give the agent a +1 reward when it is in the goal region, and 0 otherwise. Note that this RL baseline, which is similar to vanilla RL from sparse rewards, "observes" the event, providing it with additional information, while our model only uses the event probabilities learned from the

Figure 5: Visualizations of the Pusher, Maze, and Ant tasks. In the Maze and Ant tasks, the agent seeks to reach a pre-specified goal position. In the Pusher task, the agent seeks to place a block at the goal position.

success examples and receives no other supervision. It is included to provide a reference point on the difficulty of the tasks. Results are summarized in Table 2, and detailed learning curves can be seen in Figure 6 and Appendix G. We note the following salient points from these experiments.

**VICE outperforms naïve classifier.** We observe that for *Maze*, both the simple classifier and our method (VICE) perform well, though VICE achieves lower final distance. In the *Ant* environment, VICE is crucial for obtaining good performance, and the simple classifier fails to solve the task. Similarly, for the *Pusher* task, VICE significantly outperforms the classifier (which fails to solve the task). Unlike the naïve classifier approach, VICE actively integrates negative examples from the current policy into the learning process, and appropriately models the event probabilities together with the dynamical properties of the task, analogously to IRL.

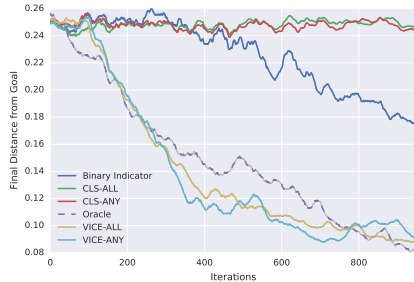

**Shaping effect of VICE.** For the more difficult ant and pusher domains, VICE actually outperforms RL with the true event indicators. We

Figure 6: Results on the Pusher task (lower is better), averaged across five random seeds. VICE significantly outperforms the naive classifier and true binary event indicators. Further, the performance is comparable to learning from an oracle hand-engineered reward (denoted in dashed lines). Curves for the Ant and Maze tasks can be seen in Appendix G.

analyze this shaping effect further in Figure 6: our framework obtains performance that is superior to learning with true event indicators, while requiring much weaker supervision. This indicates that the event probability distribution learned by our method has a reward-shaping effect, which greatly simplifies the policy search process. We further compare our method against a hand-engineered shaped reward, depicted in dashed lines in Figure 6. The engineered reward is given by $-0.2 * \|x_{block} - x_{arm}\| - \|x_{block} - x_{goal}\|$, and is impossible to compute when we don't have access to $x_{block}$, which is usually the case when learning in the real world. We observe that our method achieves performance that is comparable to this engineered reward, indicating that our automated shaping effect is comparable to hand-engineered shaped rewards.

# 7 Conclusion

In this paper, we described how the connection between control and inference can be extended to derive a reinforcement learning framework that dispenses with the conventional notion of rewards, and replaces them with events. Events have associated probabilities. which can either be provided

by the user, or learned from data. Recasting reinforcement learning into the event-based framework allows us to express various goals as different inference queries in the corresponding graphical model. The case where we learn event probabilities corresponds to a generalization of IRL where rather than assuming access to expert demonstrations, we assume access to states and actions where an event occurs. IRL corresponds to the case where we assume the event happens at every timestep, and we extend this notion to alternate graphical model queries where events may happen at a single timestep.

## Acknowledgements

This research was supported by an ONR Young Investigator Program award, the National Science Foundation through IIS-1651843, IIS-1614653, and IIS-1700696, Berkeley DeepDrive, and donations from Google, Amazon, and NVIDIA.

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
