[Supplementary Material]

# Appendices

## A  Message Passing Updates for Reinforcement Learning

In this section, we derive message passing updates that can be used to obtain an optimal policy in the graphical model for control (visualized below).

We define two backward messages, a state-action message $\beta(s_t, a_t) = p(e_{t:T} = 1|s_t, a_t)$ and a state message $\beta(s_t) = p(e_{t:T} = 1|s_t)$. The state message can be expanded in terms of the state-action message as:

$$\beta(s_t) = p(e_{t:T} = 1|s_t) = \int_{\mathcal{A}} \beta(s_t, a_t)p(a_t|s_t)da_t$$

We can then write a recursive form for the state-action message in terms of the state message:

$$\beta(s_t, a_t) = p(e_{t:T} = 1|s_t, a_t) = p(e_t = 1|s_t, a_t)p(e_{t+1:T} = 1|s_t, a_t)$$

$$= p(e_t = 1|s_t, a_t) \int_{\mathcal{S},\mathcal{A}} \beta(s_{t+1}, a_{t+1})p(a_{t+1}|s_{t+1})p(s_{t+1}|s_t, a_t)ds_{t+1}da_{t+1}$$

$$= p(e_t = 1|s_t, a_t)E_{s_{t+1}}[\beta(s_{t+1})]$$

Next, we define $\log p(e_t = 1|s_t, a_t) = r(s_t, a_t)$ as the reward factor and set the reference policy $p(a_t|s_t) = C$ to the uniform distribution as before. Non-uniform reference policies correspond to policy optimization with a modified reward function $r^{new}(s, a) = r^{old}(s, a) + \log C - \log p(a_t|s_t)$ and with a uniform reference policy. We can now assign familiar names to these messages, by defining $Q(s, a) = \log \beta(s, a)$ and $V(s) = \log \beta(s) - \log C$. Our message passing updates now resemble soft variants of Bellman backup equations:

$$V(s_t) = \log \int_{\mathcal{A}} \exp\{Q(s_t, a_t)\}da_t$$

$$Q(s_t, a_t) = [r(s_t, a_t) + \log C] + \log E_{s_{t+1}}[\exp\{V(s_{t+1})\}]$$

The constant $\log C$ term can be absorbed into the reward function to exactly match the equations we presented in Section 3.1, but we leave the term explicit for clarity of explanation. For the fixed horizon task we presented, adding a constant offset to the reward cannot change the optimal policy. As previously mentioned in Section B, under deterministic dynamics, $Q(s_t, a_t) = r(s_t, a_t) + V(s_{t+1})$, which aligns with MaxCausalEnt (Ziebart, 2010) and soft Q-learning (Haarnoja et al., 2017; Nachum et al., 2017).

From these value functions, we can easily obtain the optimal policy $p(a_t|s_t, e_{1:T} = 1)$. First note that due to conditional independence, $p(a_t|s_t, e_{1:T} = 1) = p(a_t|s_t, e_{t:T} = 1)$. Applying Bayes' rule, we now have:

$$p(a_t|s_t, e_{t:T} = 1) = \frac{p(e_{t:T} = 1|s_t, a_t)p(a_t|s_t)}{p(e_{t:T} = 1|s_t)} = \frac{\beta(s_t, a_t)C}{\beta(s_t)} = \exp\{Q(s_t, a_t) - V(s_t)\}$$

# B Control as Variational Inference

Performing inference directly in the graphical model for control produces solutions that are optimistic with respect to stochastic dynamics, and produces risk-seeking behavior. This is because posterior inference is not constrained to force $p(s_{t+1}|s_t, a_t, e_{1:T}) = p(s_{t+1}|s_t, a_t)$: that is, it assumes that, like the action distribution, the next state distribution will "conspire" to make positive outcomes more likely. Prior work has sought to address this issue via the framework of causal entropy Ziebart (2010). To provide a more unified treatment of control as inference, we instead present a variational inference derivation that also addresses this problem. When conditioning the graphical model in Figure 3 on $e_{1:T} = 1$ as before, the optimal trajectory distribution is

$$p(\tau|e_{1:T}) \propto p(s_1) \prod_{t=1}^{T-1} p(s_{t+1}|s_t, a_t)p(a_t|s_t)e^{r(s,a)}.$$

We will assume that the action prior $p(a_t|s_t)$ is uniform without loss of generality, since non-uniform distributions can be absorbed into the reward term $e^{r(s,a)}$, as discussed in Appendix A.

The correct maximum entropy reinforcement learning objective emerges when performing variational inference in this model, with a variational distribution of the form $q_\theta(\tau) = p(s_1) \prod_{t=1}^{T-1} p(s_{t+1}|s_t, a_t)q_\theta(a_t|s_t)$. In this distribution, the initial state distribution and dynamics are forced to be equal to the true dynamics, and only the action conditional $q_\theta(a_t|s_t)$, which corresponds to the policy, is allowed to vary. Writing out the variational objective and simplifying, we get

$$-D_{KL}(q_\theta(\tau)||p(\tau|e_{1:T})) = E_{\tau \sim q(\tau)} \left[ \sum_{t=1}^{T} r(s_t, a_t) - \log q_\theta(a|s) \right].$$

We see that we obtain the same problem as (undiscounted) entropy-regularized reinforcement learning, where $q_\theta(a|s)$ serves as the policy. For more in-depth discussion, see Appendix D.1. We can recover the discounted objective by modifying the dynamics such that the agent has a $(1 - \gamma)$ probability of transitioning into an absorbing state with 0 reward.

We have thus derived how maximum entropy reinforcement learning can be recovered by applying variational inference with a specific choice of variational distribution to the graphical model for control.

# C Derivations for Event-based Message Passing Updates

## C.1 ALL query

The goal of the ALL query is to trigger an event at every timestep. Mathematically, we want trajectories such that $e_{1:T} = 1$. As the ALL query is mathematically identical to MaxEnt RL, we redirect the reader to Appendix A for the derivation.

## C.2 ANY query

The goal of the ANY query is to trigger an event at least once. Mathematically, we want trajectories such that $e_1 = 1$ or $e_2 = 1 ... e_T = 1$.

First, we introduce a more concise notation by introducing a stopping time $t^* = \operatorname{argmin}_{t>0}\{e_t = 1\}$ which denotes the first time that an event happens. Asking for the stopping time to be within a certain interval is the same as asking the event to happen at least once within that interval:

$$p(t^* \in [t, T]) = p(e_t = 1 \text{ or } e_{t+1} = 1 ... e_T = 1)$$

We can now derive the message passing updates. We derive the state messages as:

$$\beta(s_t) = p(t^* \in [t, T]|s_t) = \int_{\mathcal{A}} p(t^* \in [t, T]|s_t, a_t)p(a_t|s_t)da_t$$

The state-action message can be derived as:

$$\beta(s_t, a_t) = p(t^* \in [t, T] | s_t, a_t)$$
$$= p(e_t = 1 | s_t, a_t) + p(t^* \in [t+1, T] | s_t, a_t) - p(e_t = 1 | s_t, a_t) p(t^* \in [t+1, T] | s_t, a_t)$$
$$= p(e_t = 1 | s_t, a_t) + p(e_t = 0 | s_t, a_t) p(t^* \in [t+1, T] | s_t, a_t)$$
$$= p(e_t = 1 | s_t, a_t) + p(e_t = 0 | s_t, a_t) \int_{\mathcal{S}, \mathcal{A}} p(t^* \in [t+1, T], s_{t+1}, a_{t+1} | s_t, a_t) ds_{t+1} da_{t+1}$$
$$= p(e_t = 1 | s_t, a_t) + p(e_t = 0 | s_t, a_t) E_{s_{t+1}} \left[ \int_{\mathcal{A}} p(t^* \in [t+1, T] | s_{t+1}, a_{t+1}) p(a_{t+1} | s_{t+1}) da_{t+1} \right]$$
$$= p(e_t = 1 | s_t, a_t) + p(e_t = 0 | s_t, a_t) E_{s_{t+1}} [\beta(s_{t+1})]$$

We can now define our Q and value functions as log-messages as done in Appendix A to obtain the following backup rules:

$$V(s_t) = \log \int_{\mathcal{A}} \exp\{Q(s_t, a_t)\} da_t$$

$$Q(s_t, a_t) = \log\left(p(e_t = 1 | s_t, a_t) + p(e_t = 0 | s_t, a_t) E_{s_{t+1}} [\exp\{V(s_t)\}]\right)$$

One caveat here is that the policy, $p(a_t | s_t, t^* \in [t, T])$, always seeks to make the event happening in the future, which we refer to as the *seeking* policy. The correct non-seeking policy would be indifferent to actions after the event has happened. However, in terms of achieving the objective, both policies will behave exactly the same until the event is triggered, after which the behavior of the policy will no longer matter. For example, if we operate in the first exit scenario, and consider the episode terminated after the goal event is achieved, then we never encounter the scenario when the event occurs in the past.

If we would like to compute the non-seeking policy, we can compute a forward pass which keeps track of the probability that the event has happened:

$$p(t^* \in [1, t] | s_{1:t}, a_{1:t}) = p(e_t = 1 | s_t, a_t) + p(e_t = 0 | s_t, a_t) p(t^* \in [1, t-1] | s_{1:t-1}, a_{1:t-1})$$

We can then use this forward message in conjunction with our backward messages to obtain a non-seeking policy as:

$$p(a_t | s_{1:t}, a_{1:t-1}, t^* \in [1, T]) = \frac{p(a_t | s_{1:t}, a_{1:t}, t^* \in [1, T]) p(a_t | s_t)}{p(a_t | s_{1:t}, a_{1:t-1}, t^* \in [1, T])}$$

Where

$$p(t^* \in [1, T] | s_{1:t}, a_{1:t}) = p(t^* \in [1, t-1] | s_{1:t-1}, a_{1:t-1}) + p(t^* \notin [1, t-1] | s_{1:t-1}, a_{1:t-1}) p(t \in [t, T] | s_t, a_t)$$

$$p(t^* \in [1, T] | s_{1:t}, a_{1:t-1}) = \int_{\mathcal{A}} p(t^* \in [1, T] | s_{1:t}, a_{1:t}) p(a_t | s_t) da_t$$

Note that while the policy is conditioned on all past states and actions, it only depends on them through the forward message, or the cumulative probability that the event has happened.

## D  Derivations for Variational Objectives

### D.1  ALL query

We briefly reviewed the variational derivation for standard RL in Section B. In this section, we present a more thorough derivation under the events framework and additionally discuss extensions to discounted formulations.

First, we write down the joint trajectory-event distribution, which is simply the product of all factors in the graphical model:

$$p(\tau, e_{1:T} = 1) = p(s_1) \prod_{t=1}^{T-1} p(s_{t+1} | s_t, a_t) p(a_t | s_t) p(e_t = 1 | s_t, a_t)$$

We can obtain the optimal trajectory distribution by conditioning and setting the reference policy $p(a_t|s_t)$ as the uniform distribution:

$$p(\tau|e_{1:T} = 1) \propto p(s_1) \prod_{t=1}^{T-1} p(s_{t+1}|s_t, a_t) p(e_t = 1|s_t, a_t)$$

We now perform variational inference with a distribution of the following form, where the dynamics have been forced to equal the true dynamics of the MDP:

$$q_\theta(\tau) = p(s_1) \prod_{t=1}^{T-1} p(s_{t+1}|s_t, a_t) q_\theta(a_t|s_t)$$

Here, $q_\theta(a_t|s_t)$ is the only term that is allowed to vary, and represents the learned policy. When we minimize the KL divergence between $q$ and $p$, the dynamics terms cancel and we recover the following entropy-regularized policy objective:

$$-D_{KL}(q_\theta(\tau)||p(\tau|e_{1:T} = 1)) = -E_{q_\theta}[\sum_{t=0}^{T} \log q_\theta(a_t|s_t) - \sum_{t=0}^{T} \log p(e_t = 1|s_t, a_t)] + C$$

$$= E_{q_\theta}[\sum_{t=0}^{T} \log p(e_t = 1|s_t, a_t) + H(\pi(\cdot|s_t))] + C$$

The constant C is due to proportionality in the optimal trajectory distribution, and can be ignored in the optimization process.

If we define the empirical returns $\hat{Q}$ as $\hat{Q}(s_t, a_t) = \sum_{t'=t}^{T} \log p(e_{t'} = 1|s_{t'}, a_{t'})$, we can write the returns recursively as:

$$\hat{Q}(s_t, a_t) = \log p(e_t = 1|s_t, a_t) + \hat{Q}(s_{t+1}, a_{t+1})$$

In this discounted case, we consider the case when the dynamics has a $(1 - \gamma)$ chance of transitioning into an absorbing state with reward or $\log p(e_t = 1|s_t, a_t) = 0$. This means we now adjust the recursion as:

$$\hat{Q}(s_t, a_t) = \log p(e_t = 1|s_t, a_t) + \gamma\hat{Q}(s_{t+1}, a_{t+1})$$

## D.2 ANY query

As with our derivation in the RL case, we begin by writing down our trajectory distribution. Our target trajectory distribution is $p(\tau|t^* \in [1, T])$, or trajectories where the event happens at least once.

First, we can use Bayes' rule to obtain:

$$\log p(\tau|t^* \in [1, T]) = \log p(t^* \in [1, T]|\tau) + \log p(\tau) - \log p(t^* \in [1, T])$$

The last term is a proportionality constant with respect to the trajectories. The second term is the trajectory distribution induced by the reference policy. The first term can be simplified further.

Note that the probability that the event first happens at $t^*$ is $p(t^* = t|\tau) = p(e_t = 1|s_t, a_t) \prod_{t'=1}^{t-1} p(e_{t'} = 0|s_{t'}, a_{t'})$ (i.e. the event happens at $t^*$ but not before). Now we can write:

$$p(t^* \in [1, T]|\tau) = \sum_{t=1}^{T} p(t^* = t|\tau) = \sum_{t=1}^{T} p(e_t = 1|s_t, a_t) \prod_{t'=1}^{t-1} p(e_{t'} = 0|s_{t'}, a_{t'})$$

To write down a recursion, we now define the quantity $\hat{\beta}(s_{t:T}, a_{t:T}) = p(t^* \in [t, T]|s_{t:T}, a_{t:T})$. We can now express the above term recursively as:

$$\sum_{t=1}^{T} p(e_t = 1|s_t, a_t) \prod_{t'=1}^{t-1} p(e_{t'} = 0|s_{t'}, a_{t'}) = p(e_1 = 1|s_1, a_1) + p(e_1 = 0|s_1, a_1)\hat{\beta}(s_{2:T}, a_{2:T})$$

$$= \hat{\beta}(s_{1:T}, a_{1:T})$$

Thus, if we define our empirical Q-function $\hat{Q}(s_t, a_t) = \log \hat{\beta}(s_{1:T}, a_{1:T})$, our recursion now becomes:

$$\hat{Q}(s_t, a_t) = \log(p(e_t = 1|s_t, a_t) + p(e_t = 0|s_t, a_t)e^{\hat{Q}(s_{t+1}, a_{t+1})})$$

Using the same variational distribution $q_\theta(\tau) = p(s_1) \prod_{t=1}^{T} p(s_{t+1}|s_t, a_t)q_\theta(a_t|s_t)$ as before, we can write our optimization objective as:

$$-D_{KL}(q_\theta(\tau)||p(\tau|t^* \in [1, T])) = E_q[\hat{Q}(s_{1:T}, a_{1:T}) - \sum_{t=1}^{T} \log q_\theta(a_t|s_t)] + C$$

Where the constant $C$ absorbs terms from the reference policy $p(a_t|s_t)$ which we set to uniform, and the proportionality constant $\log p(t^* \in [1, T])$.

To achieve a discounted objective case, we consider the case when the dynamics has a $(1 - \gamma)$ chance of transitioning into an absorbing state where the event can never happen $p(e_t = 1|s_t, a_t) = 0$. Note that this is different from the all query. This means we now adjust the recursion as:

$$\hat{Q}(s_t, a_t) = \gamma \log \left( p(e_t = 1|s_t, a_t) + p(e_t = 0|s_t, a_t)e^{\hat{Q}(s_{t+1}, a_{t+1})} \right) + (1 - \gamma) \log p(e_t = 1|s_t, a_t)$$

# E  Policy Gradients for Events

Because the ALL query is mathematically identical to standard RL, we do not derive the policy gradient estimator here.

For the ANY query, we consider the objective

$$J(\pi) = E_\pi[\hat{Q}(s_{1:T}, a_{1:T}) - \sum_{t=1}^{T} \log \pi(a_t|s_t)]$$

. For simplicity we disregard the entropy term as that portion remains unchanged from standard RL.

Applying logarithmic differentiation, and simplifying, we can obtain the gradient estimator.

$$E_\pi[\sum_{t=1}^{T} \nabla \log \pi(a_t|s_t)(\hat{Q}(s_{1:T}, a_{1:T}) - \log \pi(a_t|s_t))]$$

The next step is that we wish to only consider future returns, i.e. we wish to replace $\hat{Q}(s_{1:T}, a_{1:T})$ with $\hat{Q}(s_{t:T}, a_{t:T})$. First, note that before the event happens before $t$, then $\hat{Q}(s_{1:T}, a_{1:T})$ and $\hat{Q}(s_{t:T}, a_{t:T})$ are identical, but if $t$ is after then event then the returns estimator should be 0. Thus, we need to keep track of the cumulative probability that an event occurs and rewrite the estimator as:

$$E_\pi[\sum_{t=1}^{T} \nabla \log \pi(a_t|s_t)p(t \leq t^*|s_{1:t}, a_{1:t})(\hat{Q}(s_{t:T}, a_{t:T}) - \log \pi(a_t|s_t))]$$

# F  Variational Inverse Control with Events (VICE)

In this section, we explicitly write down the MLE objective for the inverse formulation of each query type (AT, ALL, ANY).

We then show that we can train a sampler for the model by optimizing a trajectory-based objective corresponding to the inference procedures outlined in Appendix D. The statement we show for each query type is that the KL between trajectory distributions upper bounds the KL between our sampler and the model we wish to draw samples from.

## F.1  AT query VICE

In the AT query, we assume we observe states and actions where the event occurred at a specific timestep, denoted as $t$. We assume our data comes from the distribution $p_{data}(s_t, a_t|e_t = 1)$

The maximum likelihood objective is:

$$\mathcal{L}_{AT}(\theta) = -E_{p_{data}}\left[\log p_\theta(s_t, a_t | e_t = 1)\right]$$

We now derive the objective for training our sampler $q(s_t, a_t)$ so that it matches $p_\theta(s_t, a_t)$. By the chain rule for KL divergence, we have the upper-bound $D_{KL}(q(s_t, a_t)||p_\theta(s_t, a_t | e_t = 1)) \leq D_{KL}(q(\tau)||p_\theta(\tau | e_t = 1))$. After obtaining $q(\tau)$, we can sample states and actions by executing full trajectories and picking the states and actions that correspond to timestep $t$.

## F.2 ALL query VICE

In the ALL query, we assume our data comes from the average distribution of states and actions along trajectories where the event happens at all timesteps (averaged over timesteps) $p_{data}(s, a | e_{1:T} = 1) = \frac{1}{T}\sum_{t=1}^{T} p_{data}(s_t, a_t | e_{1:T} = 1)$. This is similar to matching the occupancy measure of a policy, which is equivalent to inverse reinforcement learning as shown by Ho & Ermon (2016).

The maximum likelihood objective is:

$$\mathcal{L}_{ALL}(\theta) = -E_{p_{data}}\left[\log p_\theta(s_t, a_t | e_{1:T} = 1)\right]$$

We can upper-bound the KL-divergence of interest between the sampler and the model with a KL-divergence on trajectories as:

$$D(\frac{1}{T}\sum_t q(s_t, a_t)||\frac{1}{T}\sum_t p_\theta(s_t, a_t | e_{1:T} = 1))$$

$$\leq \frac{1}{T}\sum_t D(q(s_t, a_t)||p_\theta(s_t, a_t | e_{1:T} = 1))$$

$$\leq \frac{1}{T}\sum_t D(q(\tau)||p_\theta(\tau | e_{1:T} = 1))$$

$$= D(q(\tau)||p_\theta(\tau | e_{1:T} = 1))$$

The first inequality comes from the log-sum inequality, and the second inequality comes from the chain rule for KL divergence.

## F.3 ANY query VICE

In the ANY query formulation, we assume our data comes from the distribution of states and actions at the first timestep an event happens, $p_{data}(s_{t^*}, a_{t^*} | t^* \in [1, T])$.

$$\mathcal{L}_{ANY}(\theta) = -E_{p_{data}}\left[\log p_\theta(p_{data}(s_t, a_t | t^* = t))\right]$$

To show that optimizing the trajectory distribution bounds, we first rewrite $p(s_{t^*}, a_{t^*} | t^* \in [1, T])$ over timesteps as $p(s_{t^*}, a_{t^*} | t^* \in [1, T]) = \sum_{t=1}^{T} p(s_t, a_t | t^* = t)p(t^* = t)$.

**Lemma F.1.** *Let $X = (x_1, x_2, ...)$, $Y = (y_1, y_2, ...)$. Let $\bar{\mu}$ denote a set of weights which sum to one, and denote $\bar{p}(X) = E_{\bar{\mu}}[p_i(x_i)]$, and $\bar{p}(X, Y) = E_{\bar{\mu}}[p_i(x_i, y_i)]$ denote convex combinations of the individual distributions $p_i$. Then,*

$$D(\bar{p}(X, Y)||\bar{q}(X, Y)) \geq D(\bar{p}(X)||\bar{q}(X))$$

*Proof.* This statement directly follows from the chain rule for KL divergences, which implies:

$$D(\bar{p}(\mathbf{X}, \mathbf{Y})||\bar{q}(\mathbf{X}, \mathbf{Y})) = D(\bar{p}(\mathbf{X})||\bar{q}(\mathbf{X})) + D(\bar{p}(\mathbf{X}|\mathbf{Y})||\bar{q}(\mathbf{X}|\mathbf{Y})) \geq D(\bar{p}(\mathbf{X})||\bar{q}(\mathbf{X}))$$

$\square$

Now, we can apply Lemma F.1 to derive the upper-bound:

$$D(\sum_t q(s_t, a_t)p(t^* = t) || \sum_t p_\theta(s_t, a_t | t^* = t)p(t^* = t))$$

$$\leq D(\sum_t q(\tau)p(t^* = t) || \sum_t p_\theta(\tau | t^* = t)p(t^* = t))$$

$$= D(q(\tau) || p_\theta(\tau | t^* \in [1, T]))$$

We can obtain samples from $q(s_{t^*}, a_{t^*})$ by executing full trajectories and using the first state when an event is triggered.

### F.4 Justification for using the discriminator

In the previous section, we have justified the algorithm which updates the model via the gradient Eqn. 1, by training a sampling policy that minimizes KL to the model distribution.

In Section 5.1, we propose to implement the update via training a discriminator instead of an energy-based model $p_\theta(s, a | e = 1)$. An approximate connection can be made in this case, which ignores changes in the state-distribution of the sampling policy. To see this, we represent the state-action marginal of the policy as $q(s, a) = q(a|s)\bar{p}(s)$, where $\bar{p}(s)$ is the state-marginal of the reference policy (set to uniform, see Appendix A). Note that this is not the state distribution induced by the policy, $q(s)$.

We can use Bayes rule to write our model as $p_\theta(s, a|e = 1) \propto p_\theta(e = 1|s, a)\bar{p}(a|s)\bar{p}(s)$, meaning our model is only parameterized by the event probability.

Following previous work Finn et al. (2016b), we model the discriminator as $D_\theta(s, a) = \frac{p_\theta(s,a|e=1)}{p_\theta(s,a|e=1)+q(s,a)} = \frac{p_\theta(e=1|s,a)+C_\theta}{p_\theta(e=1|s,a)+C_\theta+q(a|s)}$, where $C_\theta$ is a learnable constant that corresponds to proportionality factors.

The inconsistency with using $q(s, a) = q(a|s)\bar{p}(s)$ instead of $q(s, a) = q(a|s)q(s)$ arises in the policy optimization objective, which is minimizing the KL between the latter quantity an the model. This means that we do not draw unbiased negative examples for training the discriminator, which is also noted in (Fu et al., 2018).

## G Experiments

### G.1 Experimental details for prespecified events

On the *Lobber* task, we use a diagonal gaussian policy where the mean is parametrized by a 32x32 neural network. We use a TRPO batch size of 40000 and train for 1000 iterations.

On the *HalfCheetah* task, we use a diagonal gaussian policy where the mean is parametrized by a 32x32 neural network. We use a TRPO batch size of 10000 and train for 1000 iterations.

### G.2 Experimental details for learning event probabilities

We evaluate the performance of VICE in learning event probabilities on the *Ant*,*Maze*, and *Pusher* tasks, providing comparisons to classifier-based methods. Although the binary indicator baseline is not comparable to VICE (since it observes the event while the other methods do not), we present comparisons to provide a general idea of the difficulty of the task. All experiments are run with five random seeds, and mean results are presented.

We use Gaussian policies, where the mean is parametrized by a neural network, and the covariance a learned diagonal matrix. The event distribution is represented by a neural network as well. Further hyperparameters are presented in Table 3.

On the *Ant* task, both the policy mean network and event distribution network have two hidden layers with 200 units and ReLu activations.

On the *Maze* task, the mean network has two convolutional layers, with filter size $5 \times 5$, followed by two fully connected layers with 32 units each with ReLu activations. The event distribution is

represented using a convolutional neural network with two convolutional layers with $5 \times 5$ filters, and a final fully-connected layer with 16 units.

On the *Pusher* task, the policy is represented by a convolutional neural network with three convolutional layers, with a stride of 2 in the first layer, and a stride of 1 in the subsequent layers. We use a filter size of 3x3 in all the layers, and the number of filters are 64, 32 and 16. In line with prior work (Finn et al., 2016a), we pre-train the convolutional layers using an auto-encoder loss on data collected from random policies. The fully connected part of the neural network consists of two layers, each having 200 units and ReLu activations to represent the policy. The event distribution is also represented by the same architecture.

|  | Ant | Maze | Pusher |
|---|---|---|---|
| Batch Size | 10000 | 5000 | 10000 |
| Iterations | 1000 | 150 | 1000 |
| Discount | 0.99 | 0.99 | 0.99 |
| Entropy | 0.1 | 0.1 | 0.01 |
| # Demonstrations | 500 | 1000 | 10000 |

Table 3: Hyperparameters used for VICE on the Ant,Maze, and Pusher tasks

(a) Ant

(b) Maze

(c) Pusher

Figure 7: Learning curves for the various methods for the *Ant*,*Maze*, and *Pusher* tasks, averaged across five random seeds. On all three domains, VICE-ALL and VICE-ANY successfully solve the task consistently, while the naive classifier fails often. Although the binary indicator works reasonably on the Maze task, it fails to solve the task in the more challenging environments.

## G.3 Detailed learning curves for learning event probabilities

(a) Ant - VICE-ALL    (b) Maze - VICE-ALL    (c) Pusher - VICE-ALL

(d) Ant - VICE-ANY    (e) Maze - VICE-ANY    (f) Pusher - VICE-ANY

(g) Ant - CLS-ALL    (h) Maze - CLS-ALL    (i) Pusher - CLS-ALL

(j) Ant - CLS-ANY    (k) Maze - CLS-ANY    (l) Pusher - CLS-ANY

(m) Ant - Binary Indicator    (n) Maze - Binary Indicator    (o) Pusher - Binary Indicator

Figure 8: Learning curves for all methods on each of the five random seeds for the *Ant*,*Maze*, and *Pusher* tasks. The mean across the five runs is depicted in bold.