[Reviews · NeurIPS 2018]

Reviewer 1



Reward functions often pose a bottleneck to real-world applications of reinforcement learning. Inverse reinforcement learning tries to overcome this problem by learning to mimic expert behavior. But this requires expert demonstrations which can be difficult or expensive in practice. This paper proposes an inverse event-based control, which generalizes inverse reinforcement learning methods to cases where full demonstrations are not needed, but only to know the probability that a certain event occurred. Authors have done a considerable amount of work. However, I feel that the presentation of the paper needs to be improved before it can be accepted. In the current form, the paper is not self-contained and it is very hard to follow. It refers to additional material several times. I feel that it will be unfair to other authors who try to make their papers self-contained despite additional materials. == In the rebuttal, the authors have agreed to improve the clarity of the paper for the final version. I have increased the score.

Reviewer 2



**Summary of the paper The paper considers a problem of learning a policy that reaches specified goal states (known as events) without an access to the true reward function. The paper proposes a method that alternates between learning a reward function and learning a policy. Algorithmically, the proposed method resembles inverse reinforcement learning/imitation learning. However, unlike existing methods that requires expert trajectories, the proposed method only requires goal states that the expert aims to reach. Experiments show that the proposed method reaches the goal states more accurately than an RL method with a naïve binary classification reward. **Detailed comments: The paper extends the problem setting of inverse reinforcement learning and proposes a novel method to solve it. The proposed method is interesting, and I enjoy reading the paper. However, I have a concern regarding a gap between the theorems and the proposed practical algorithm. - By using logistic regression to classify an event label given a state-action pair, the classifier D(s,a) is an estimate of a positive class (event happens) posterior p(e=1|s,a), and 1 – D(s,a) is an estimate of a negative class (event does not happen) posterior p(e=-1|s,a). However, in line 6 of Algorithm 1, both positive and negative class posteriors are used to update \hat{p}(e=1|s,a). This is strange unless log \hat{p}(e=1|s,a) should be understood as a reward function and not as a log of class posterior. For such a case, this choice of reward function seems fine since minimizing log(1-D(s,a)) implies maximizing log(D(s,a)). However, it is not consistent with the theorems. Specifically, the reward function for the Q-value in Theorem 1 is the log of positive class posterior and not the difference between the log posteriors of the two classes which is used in Algorithm 1. I think that this is a big gap between theory and practice, and should be explained better in the paper. **Minor comments - In line 247, the left side of the relation should be conditioned on the event e_t=1. This also applied to the subsequent gradient equation. - In the IRL method of Fu et al., 2018, the discriminator depends on policy, D(s,a) = f(s,a) / (f(s,a) + pi(a|s)). It is unclear in the paper whether it uses this policy dependent discriminator or the policy independent one used by GAIL and GAN. ======================================================= I read the rebuttal letter and have a better understanding of the method. I am more confidence on the paper and increase the score.

Reviewer 3



The paper presents a new method of RL and draws the connection to previous work very well. I am not an expert in all of the presented methods, which is why I cannot clearly asses the novelty of the presented approach. The results are convincing.